# Evaluation of Electric Vehicle Integrated Charging Safety State Based on Fuzzy Neural Network

**Hui Gao [1], Binbin Zang [1,\*], Lei Sun [2] and Liangliang Chen [3]**

[1] College of Automation, College of Artificial Intelligence, Nanjing University of Posts and Telecommunications, Nanjing 210023, China; gaohui2005@163.com
[2] State Grid Jiangsu Electric Power Co., Ltd., Research Institute, Nanjing 211103, China; slsz_2014@126.com
[3] NARI Technology Co., Ltd., Nanjing 211106, China; N1220055918@126.com
[\*] Correspondence: nyzangbinbin@163.com

**Featured Application: This work is suggested to be applied to the safety assessment of electric vehicle charging process, which may be embedded into the system to effectively assess the charging state and provide certain help for reducing charging failures in the future.**

**Abstract:** Electric vehicles have been promoted worldwide because of their high energy efficiency and low pollution. However, frequent charging safety accidents have to a certain extent restricted the development of electric vehicles. Therefore, it is extremely important to accurately evaluate the safety state of EV charging. The paper presents an integrated safety assessment method for electric vehicle charging safety based on fuzzy neural network. The integrated fault model was established by analyzing the correlation between truck–pile–grid. Then the integrated evaluation index was analyzed and sorted out, and the comprehensive fuzzy evaluation method used to evaluate. Following this, the improved GA_BP neural network algorithm was used to calculate the weight. Compared with the evaluation effect before and after the improvement, the simulation results show that the GA_BP neural network has higher accuracy and smaller error than the ordinary BP neural network. Finally, the feasibility and effectiveness of the evaluation method was verified by a case study.

**Keywords:** electric vehicles; fuzzy neural network; safety assessment method; evaluating indicator

## 1. Introduction

In order to promote a global carbon balance, countries have formulated their own "carbon reduction" targets [1]. China is striving to achieve the "carbon peak" before 2030 and the "carbon neutral" "dual carbon" goal before 2060 [2]. New energy electric vehicles play an important role in reducing carbon emissions, reducing consumption of fossil energy, and promoting the development of electrified transportation [3]. However, in the promotion process of electric vehicles, power battery failure and charging equipment safety problems have become obstacles to the rapid development of electric vehicles [4]. Therefore, EV fault diagnosis, safety warnings, and other issues have become the focus of research in various countries [5].

As the main energy source of the vehicle, the power battery system is not only the core component of electric vehicles, but also a technical bottleneck restricting their development [6]. The inevitable performance degradation of the power battery in use will lead to the decrease of the vehicle driving range, the deterioration of the power performance and the shortening of service life, which will cause an increase in the safety risks. Accurate prediction and diagnosis of power battery faults is an important guarantee to improve the safety and reliability of electric vehicles [7]. In the actual operation of electric vehicles, many factors such as electromagnetic interference, road conditions, and driving habits can lead to battery system failures, and complex, nonlinear, or multi-parameter

coupling. First, voltage difference, covariance matrix, and variance matrix are used as input values of the generalized regression neural network to classify fault states. Then, the wavelet neural network is used for fault detection, which can significantly improve the efficiency and the accuracy of the classification of efficiency fault degree [8]. According to the statistical analysis of big data of electric vehicles, the fault diagnosis of the battery power supply system can clarify the fault type, locate the fault, and avoid the occurrence of faults, which has certain practical value and plays a very positive role in improving the stability of electric vehicles [9]. The application of a battery fault diagnosis expert system in a battery management system is utilized to diagnose power battery faults timely and accurately to ensure the safe and reliable operation of equipment [10]. In terms of power battery safety evaluation, a lithium-ion battery health evaluation framework based on online measured parameters such as voltage, current, and critical time interval was proposed in reference [11]. By mapping degradation characteristics to capacity space, the actual capacity and health status of batteries can be estimated. The advantage is in using adaptive degradation characteristics and algorithms requiring less computing resources.

As the direct carrier of electric vehicle power supply, the charging security of the charging pile foundation needs to be properly guaranteed in order to guarantee the stable development of the electric vehicle industry [12]. Therefore, the safety of EV charging stations is extremely important. At present, the quantitative evaluation of electrical safety considering the operating conditions of large EV charging stations is still a challenge. In reference [13], the electrical safety of a large-scale electric vehicle power supply system coupled with a renewable energy power generation system was evaluated. In order to accurately locate and quickly solve the fault of an EV charging device, the fault diagnosis system of the EV charging device based on fault tree analysis can achieve rapid fault location and greatly shorten the maintenance time [14]. In order to ensure normal operation and operation of EV charging device vehicles, the cloud platform remote control system monitoring and maintenance system is used to obtain EV charging data and real-time information of faulty equipment [15].

In terms of safety assessment, the principles and methods of systems engineering are used mainly to comprehensively evaluate and predict the possible risks and possible consequences of the proposed and existing projects [16]. Because the security problems in different professional fields are very different, the research on comprehensive security assessment methods comes generally from traditional or improved comprehensive assessment methods. The evaluation methods are mainly divided into analytic hierarchy process, fuzzy evaluation, data envelopment analysis, grey correlation analysis and so on. At present, relevant studies are mostly about the health assessment of electric vehicle power batteries [17] or the safety state assessment of charging facilities [18]. There are relatively few studies on integrated safety evaluation of the power battery, charging equipment, and power supply, and no effective integrated safety evaluation index system has been developed. In this paper, by studying the integrated safety assessment system of electric vehicle charging, an early warning of the integrated charging safety of battery, charging, and power supply is derived. This will play an effective supporting role in improving the evaluation effect of the charging safety system, ensuring the safety of vehicle charging, and promoting the development of electric vehicles.

The paper is organized as follows: Section 2 analyzes the fault safety mechanism of vehicle pile and grid, and establishes the correlation between vehicle, pile, and grid; An integrated fault tree model is established based on correlation analysis. Section 3 establishes the integrated charging safety assessment model based on the fuzzy neural network algorithm. In Section 4, the algorithm is simulated, and then the feasibility and effectiveness of the proposed method is verified by case analysis. The last section concludes the article.

## 2. Fault Correlation Analysis of Vehicle–Pile–Grid Charging

The safety of electric vehicle charging [19] is a multi-layer and complex problem of the system, including the power battery, the charging equipment, the distribution network, and other factors. Currently, safety of charging is a major research direction. The existing studies on the safety of electric vehicles mainly start from the independent perspectives of power batteries [10] and the safety protection of charging equipment [20]. To study the safety of EV charging in a more comprehensive and in-depth way, this chapter integrates the three aspects of the EV power battery, charging equipment, and power grid. The safety mechanism and correlation of electric vehicle charging are analyzed, and the factors affecting charging safety are also analyzed to provide theoretical basis for subsequent research on electric vehicle charging safety assessment.

### 2.1. The Safety Mechanism of Vehicle–Pile–Grid Charging

When the EV is in the normal charging state, the EV power battery obtains electrical energy from the regional power grid through a charging station to meet charging demand [21]. When the battery fails, it will first affect the electric vehicle itself, and may directly affect the pile (burn out the power electronic protection device of the pile). If the charging pile fails, the charging safety of the EV will be affected first (may cause overvoltage, overcurrent, overcharge, spontaneous combustion, etc.), and if serial failures of the charging piles occur in some areas, it may affect the local power grid (voltage impact, voltage overlimit, reduce power quality, etc.) [22]. The probability of regional power grid failure is relatively small. If power grid failure occurs, it will directly affect the pile (overvoltage, overcurrent, harmonics, etc.), so that electric vehicles cannot be charged normally. Figure 1 shows the safety interaction mechanism of vehicle pile–network integration caused by a single fault of the power battery, charging pile, and regional power grid.

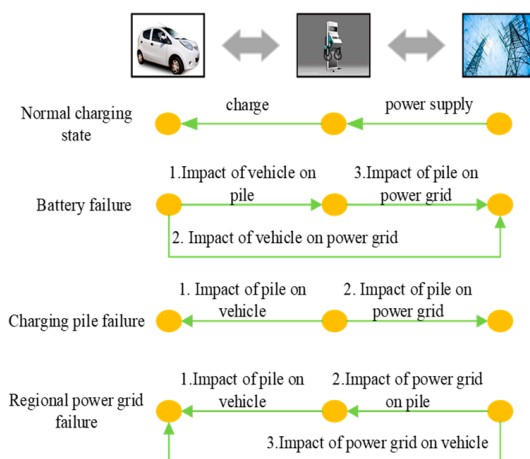

**Figure 1.** Vehicle–Pile–Grid Integrated Safety Mechanism.

### 2.2. Multistage Fault Correlation Analysis

Power battery faults mainly include battery pack capacity reduction, the charging voltage is too high, the battery pack cannot charge, a low discharge voltage, large self-discharge, local high temperature, poor voltage consistency of the single cell, battery arc breakdown, single cell damage, etc., [23]. The fault phenomena of the charging equipment mainly include a charging gun fault, an abnormal program discrimination of the charging machine, an abnormal BMS report, an internal communication fault of the charging machine, a voltage and current fault, a communication terminal fault with BMS, an electronic lock opening and closing fault, an overtemperature fault, a mechanical fault of the charging module, etc., [24]. Distribution network faults include off-line voltage, system instability, etc., [25].

An information flow model was used to analyse the correlation between faults [26]. The basic elements of an information flow model are tests, fault isolation conclusions, testable inputs, untestable inputs, and trouble-free conclusions. The model uses a directed graph to represent the causal relationship between tests and fault conclusions, in which the nodes are the test and the fault isolation conclusions, and the edges are the directions of information flow. In the information flow model, testing refers to all sources of information used to determine the health of a system, such as observed abnormal physical phenomena, abnormal data, and so on. $T_j$ is used to represent the test in the model ($j$ = 1,..., $N$, $N$ is the total number of tests in the system), all tests are binary, if $t_j$ output is normal, it is "0", otherwise, it is "1". The test is symmetric if the information obtained at "0" is equivalent to that obtained at "1". In the model, each test is "wrapped independently," meaning that each test can be carried out independently, regardless of order.

The fault isolation conclusion includes all the failures of components, the components, subsystems and systems, special non-hardware faults, special multiple faults, lack of fault indication, abnormal system input and so on. The conclusion of the fault isolation is equivalent to the conclusion of the function diagnosis of each part of the system and is related to a certain maintenance level. In the model, $f_i$ is used to represent the fault isolation conclusion ($i$ = 1,..., $M$, $M$ is the total number of fault isolation conclusions in the system). When the fault corresponding to $f_i$ occurs, $f_i$ is "1", otherwise it is "0".

The correlation information flow model was established according to the possible fault types of the vehicle–pile–grid analysed above and the information flow construction mode. $T_i$ is the fault serial number, which is divided into SOC fault, over charge, over discharge, over temperature, spontaneous combustion, communication, software, mechanical, electrical, distribution network fault, and distribution network warning, etc. Distribution network outage failure will not charge the EV distribution network, the warning being relatively isolated. The relationship is shown in Figure 2.

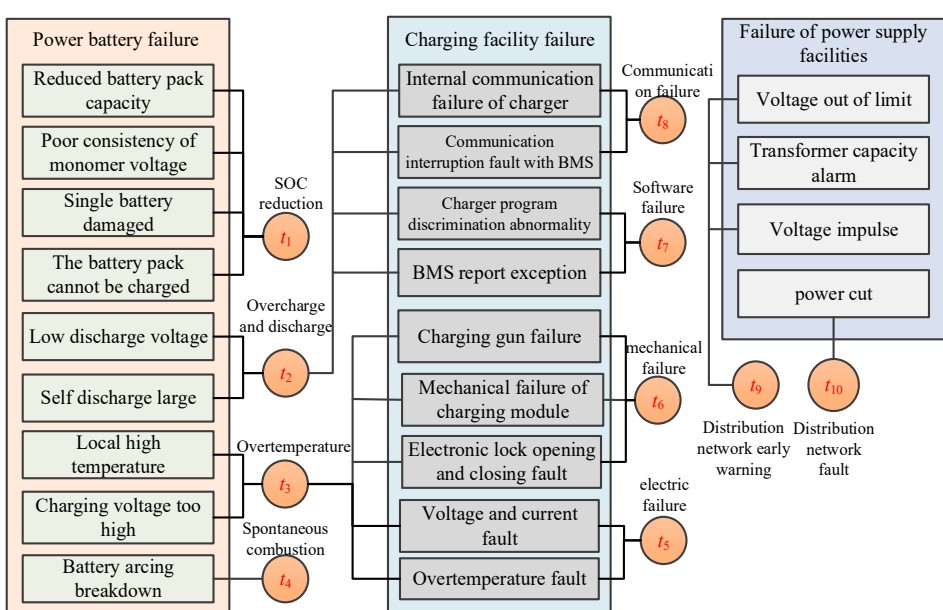

**Figure 2.** Information flow model of equipment fault correlation in the charging process.

### 2.3. Integrated Fault Tree Model Is Established

A typical accident tree structure contains at least the following components: top event (*T*), intermediate event (*M_i*), basic event (*R_i*) and logical relation (Gate) [27]. In the process of accident tree analysis, $M_i$ and $R_i$ can be expressed as

$$\left.\begin{array}{l} M_i = \{M_1, M_2, M_3, \cdots, M_n\} \\ R_i = \{R_1, R_2, R_3, \cdots, R_n\} \end{array}\right\} \tag{1}$$

According to the logic operation theory, the calculation formula of logical "and" and "or" in the accident tree is as follows:

$$\Phi(x) = \sum_{i=1}^{n} x_i = \{x_1 + x_2 + x_3 + \cdots + x_n\} \tag{2}$$

$$\Psi(x) = \prod_{i=1}^{n} x_i = \{x_1 \times x_2 \times x_3 \times \cdots \times x_n\} \tag{3}$$

In the formula, $\Phi(x)$ and $\Psi(x)$ are the events at the top of the fault tree caused by "or" doors and "and" doors respectively, and $x_i$ is the events leading to the occurrence of the events at the top.

The steps for establishing the integrated fault tree of vehicle–pile–grid are mainly divided into seven steps, which are explained as follows:

(1) Familiarity with the system: The multi-stage equipment composed of EV, charging equipment, and distribution network in the process of EV charging is regarded as a system. It is necessary to master the parameter changes of each equipment state in the charging process of multi-stage equipment.

(2) Investigating accidents: Counting the failure events of multi-level equipment in the charging history of electric vehicles, researching the causal relationship of multi-level equipment failures, and summarizing the possible failures of the system.

(3) Determine the overhead event: Identify the source of all faults in multi-stage equipment during EV charging, which is also the parent event of all multi-stage equipment faults.

(4) Determine the target value: First, analyze the multi-stage equipment failure events in the charging process of EV. Then, the probability of multistage equipment failure is obtained by statistical analysis, which is taken as the target value.

(5) Investigate causes of the incident: Investigate and analyze the specific cause of the accident of multi-stage equipment in the charging process of EV.

(6) Draw the fault tree: With the event at the top as the source, the fault events in all multi-level equipment in the charging process of EV are arranged step by step in a tree form combined with logical relations, and finally the fault events to be analyzed are ranked.

(7) Analysis: Combining the faults of the same category in the multi-stage equipment fault tree during the charging process of electric vehicles, so that it forms a subclass of root failure, simplifying the structure of the multi-stage equipment failure tree, and finally determining the importance of each basic failure event.

The integrated online fault diagnosis tree of multilevel equipment is established, including three first-level fault sources: power battery, charging facility, and power supply facility. Each primary fault source contains secondary or even tertiary fault sources and specific fault types and the power battery and charging facility faults are of the same specific fault types, such as overvoltage, overcurrent, and overtemperature. This is also a major cause of security failures. Through the analysis of the integrated fault tree, the integrated online fault diagnosis of multi-stage equipment in the charging process of electric vehicles can be realized. Figure 3 shows the fault tree of the integrated online fault diagnosis for multi-level devices. $T$ is the top event, indicating charging failure. $M_1$, $M_2$, $M_3$ indicate power battery failure, charging facility failure, and distribution network failure respectively. $M$ represents intermediate events and $R$ represents the basic event. The fault isolation conclusion includes all the failures of components, the components, subsystems and systems, special non-hardware faults, special multiple faults, lack of fault indication, abnormal system input and so on as Table 1.

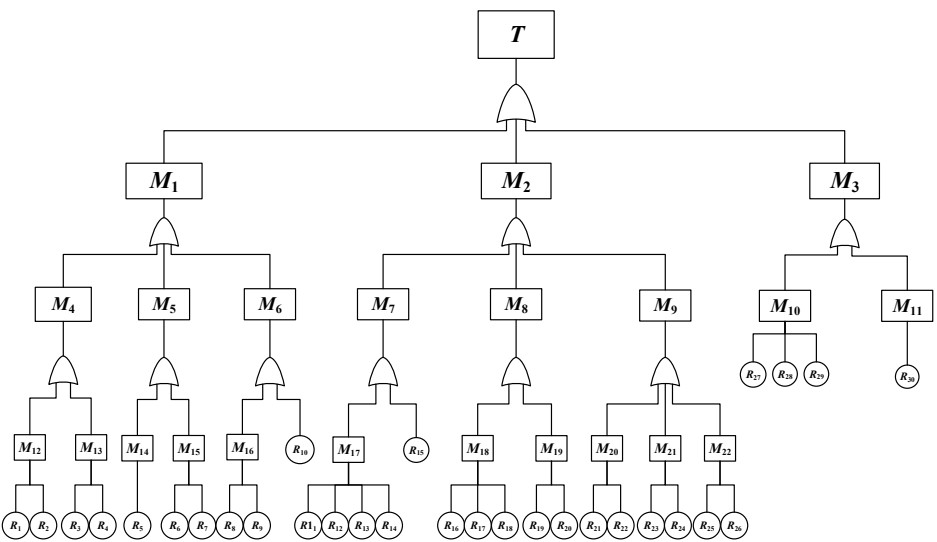

**Figure 3.** Integrated fault tree model.

**Table 1.** Fault symbols and fault types.

| Symbol | The Fault Types | Symbol | The Fault Types |
|--------|-----------------|--------|-----------------|
| $M_4$ | SOC reduction | $R_7$ | electric leakage |
| $M_5$ | Overcharge and discharge | $R_8$ | Increase of internal resistance |
| $M_6$ | Overtemperature | $R_9$ | Monomer overcharge |
| $M_7$ | mechanical failure | $R_{10}$ | Local high temperature |
| $M_8$ | Electrical fault | $R_{11}$ | Terminal breakage |
| $M_9$ | Software failure | $R_{12}$ | Cable damage |
| $M_{10}$ | Distribution network early warning | $R_{13}$ | Shell damage |
| $M_{11}$ | Distribution network fault | $R_{14}$ | Overtemperature |
| $M_{12}$ | The battery pack is not easy to charge | $R_{15}$ | Electronic lock damaged |
| $M_{13}$ | Poor consistency of single cell | $R_{16}$ | short circuit |
| $M_{14}$ | Low discharge voltage | $R_{17}$ | Overpressure |
| $M_{15}$ | High self-discharge | $R_{18}$ | Undervoltage |
| $M_{16}$ | Charging voltage too high | $R_{19}$ | Fan failure |
| $M_{17}$ | Charging gun failure | $R_{20}$ | Overpressure |
| $M_{18}$ | Voltage fault | $R_{21}$ | Relief circuit failure |
| $M_{19}$ | Overtemperature fault | $R_{22}$ | Address recognition error |
| $M_{20}$ | Abnormal charger program | $R_{23}$ | Message timeout |
| $M_{21}$ | BMS report exception | $R_{24}$ | Address error |
| $M_{22}$ | Communication failure | $R_{25}$ | Communication timeout |
| $R_1$ | Increase of internal resistance | $R_{26}$ | Communication interruption |
| $R_2$ | Internal short circuit | $R_{27}$ | Over voltage |
| $R_3$ | Increase of internal resistance | $R_{28}$ | Insufficient capacity |
| $R_4$ | Internal short circuit | $R_{29}$ | Voltage impulse |
| $R_5$ | Increase of internal resistance | $R_{30}$ | Power failure |
| $R_6$ | SOC imprecision | | |

## 3. The Integrated Charging Safety Evaluation Model Based on Fuzzy Neural Networks

### 3.1. The Fault Tree Model Is Embedded in the BP Neural Network Module

Figure 4 shows the process of embedding the fault tree into the BP neural network. Basic events obtained from the fault tree analysis are extracted and processed to obtain the basic safety evaluation factors, which are then used as the input layer neurons of the BP neural network, namely:

$$\{R_1, R_2, R_3, \cdots, R_n\} = \{x_1, x_2, x_3, \cdots, x_n\} \tag{4}$$

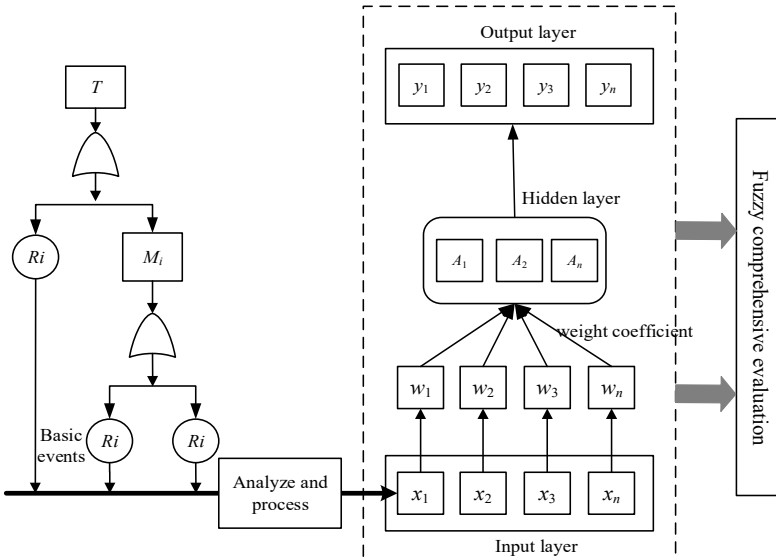

**Figure 4.** Schematic diagram of fault tree embedded BP neural network.

The evaluation indicator of event $T$ on top of the fault tree corresponds to the neurons of the output layer, which is a collection of synaptic weights in the network. The framework and diagram are as follows:

### 3.2. Fuzzy Comprehensive Evaluation

The evaluation indicator of event $T$ on top of the fault tree corresponds to the neurons of the output layer, which is a collection of synaptic weights in the network. The framework and diagram are as follows:

For the complex electric vehicle integrated charging process, there are many evaluation predictors to consider, and the factors are at different levels. The application of single-level fuzzy evaluation analysis will not produce accurate evaluation results. Therefore, evaluation factors need to be classified by attribute, divided into power batteries, charging facilities and distribution networks, each category of evaluation, and then the results of various types of assessment of a multi-level comprehensive assessment [28].

(1)　Determine the set of evaluation factors

During charging, the safety problems are divided into $n$ evaluation factors according to the predicted objects such as power batteries, charging equipment, and distribution network equipment:

$$U = \{U_i\}, \ i = 1, 2, \cdots, n \tag{5}$$

In the above, $U_i$ represents the *ith* factor in the system being evaluated. The collection of evaluation factors $U_i$ is divided into $m$ subsets by different failure types:

$$U_i = \{u_{ij}\}, \ j = 1, 2, \cdots, m \tag{6}$$

(2)　Determine the review set

In view of the safety influencing factors, the safety prediction evaluation indicator set is given, and the evaluation set is set for convenience

$$V = \{v_1, v_2, \cdots, v_s\} \tag{7}$$

where: $v_k$ indicates the result of the evaluation rating of things, $s$ represents the number of evaluation grades, and is 5 in this paper. The integrated safety evaluation system of electric vehicles is rated. $v_1$ means absolute security, $v_2$ means security, $v_3$ means general security, $v_4$ means dangerous, $v_5$ indicates very dangerous.

(3)    Single-factor fuzzy evaluation

Starting from a single factor in the factor set $U$, it is assumed that the $i$th factor $u_i$ has $m$ fault types, and the $j$th fault type is represented by $u_{ij}$. The membership of the $k$th comment level $v_k$ is $r_{jk}$, and $k$ ranges from 1 to $s$. Then the result of the $i$th evaluation factor is:

$$R_i = \begin{bmatrix} r_{11} & r_{12} & \cdots & r_{1s} \\ r_{21} & r_{22} & \cdots & r_{2s} \\ \vdots & \vdots & & \vdots \\ r_{m1} & r_{m2} & \cdots & r_{ms} \end{bmatrix} \tag{8}$$

(4)    The establishment of the weight set

In general, each fault type has a different degree of influence on the evaluation result, so each factor $u_i$ is given a corresponding weight $w_{ij}$ ($j = 1,2,3,..., m$), then the weight number of each factor constitutes the weight set $w_i$:

$$w_i = (w_{i1}, w_{i2}, \cdots, w_{im}) \tag{9}$$

In the formula, $w_i$ represents a fuzzy subset on $u_i$ and meets the criteria:

$$\sum_{j=1}^{m} w_{ij} = 1 \; and \; 0 < w_{ij} < 1 \tag{10}$$

(5)    Single-factor fuzzy comprehensive evaluation

Supposing you obtain the result of the $i$th factor fuzzy comprehensive evaluation after a fuzzy transformation:

$$B_i = w_i \circ R = (w_{i1}, w_{i2}, \cdots, w_{im}) \circ \begin{bmatrix} r_{11} & r_{12} & \cdots & r_{1s} \\ r_{21} & r_{22} & \cdots & r_{2s} \\ \vdots & \vdots & & \vdots \\ r_{m1} & r_{m2} & \cdots & r_{ms} \end{bmatrix} = (b_{i1}, b_{i2}, \cdots, b_{is}) \tag{11}$$

$B_i$ refers to the fuzzy comprehensive evaluation result of category $i$ factors, where $\circ$ is a fuzzy operator, and the safety evaluation of EV charging involves multiple indicators from multiple aspects including battery, charging equipment, and distribution network. To avoid ignoring some key information and to get comprehensive evaluation results, the weighted summation generalized fuzzy operator $M(\cdot, \oplus)$ is adopted in this paper when comprehensively considering the influence of all factors and the degree of membership of the evaluation object to each element in the evaluation set.

(6)    Multi-level fuzzy comprehensive evaluation

In this paper, a two-layer fuzzy comprehensive evaluation method is adopted. First, the two-layer single-factor fuzzy comprehensive evaluation matrix $R$ is obtained from the one-layer single-level fuzzy comprehensive evaluation matrix:

$$R = \begin{bmatrix} B_1 \\ B_2 \\ \vdots \\ B_n \end{bmatrix} = \begin{bmatrix} w_1 \circ R_1 \\ w_2 \circ R_2 \\ \vdots \\ w_n \circ R_n \end{bmatrix} = \begin{bmatrix} b_{11} & b_{12} & \cdots & b_{1s} \\ b_{21} & b_{22} & \cdots & b_{2s} \\ \vdots & \vdots & & \vdots \\ b_{n1} & b_{n2} & \cdots & b_{ns} \end{bmatrix} \tag{12}$$

The operator model $M(\cdot, \oplus)$ is adopted in the two-layer fuzzy comprehensive evaluation, and the two-layer fuzzy comprehensive evaluation matrix $B$ is:

$$B = W \times R = \begin{bmatrix} w_{11} & w_{12} & \cdots & w_{1n} \\ w_{21} & w_{22} & \cdots & w_{2n} \\ \vdots & \vdots & & \vdots \\ w_{m1} & w_{m2} & \cdots & w_{mn} \end{bmatrix} \times \begin{bmatrix} b_{11} & b_{12} & \cdots & b_{1s} \\ b_{21} & b_{22} & \cdots & b_{2s} \\ \vdots & \vdots & & \vdots \\ b_{n1} & b_{n2} & \cdots & b_{ns} \end{bmatrix} = \begin{bmatrix} c_{11} & c_{12} & \cdots & c_{1s} \\ c_{21} & c_{22} & \cdots & c_{2s} \\ \vdots & \vdots & & \vdots \\ c_{m1} & c_{m2} & \cdots & c_{ms} \end{bmatrix} \tag{13}$$

The two-layer fuzzy comprehensive judgment diagram is shown in Figure 5 below.

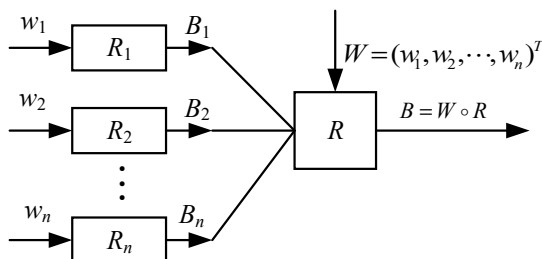

**Figure 5.** The two-layer fuzzy composite judgment diagram.

Usually, "maximum membership degree principle" is used to take the maximum number in the fuzzy matrix $B$ as the final evaluation result, which is simple, but often the processing results include only part of the information and do not fully reflect the evaluation results. In order to fully reflect the information in $B$, this topic will be the corresponding parameters of each evaluation grade, one by one, corresponding to form the evaluation parameters scoring column vector, $T = (t_1, t_2, \cdots, t_s)^T$:

$$f = B \times T \tag{14}$$

$f$ is the evaluation score of each object, which is used to judge the rating of each evaluation object.

### 3.3. Genetic Algorithms Optimize Fuzzy BP Neural Networks

It can be seen from the comprehensive fuzzy evaluation model that it is very important to determine the weight value of indexes objectively and reasonably. Therefore, a BP neural network model is established to train the weight of each index and optimize the weight accuracy [29] by learning the typical data of fuzzy comprehensive evaluation, so as to more reasonably evaluate the safety level of the integration of the vehicle-pile-network.

However, the BP neural network algorithm has uncertainty in the initial weight assignment, and the local index weight will produce a minimum value. However, GA can improve the accuracy of the network initial value [30]. Better individuals are found in the initial weight population to make the actual output as close as possible to the expected output to avoid the randomness of the weight initialization. It can prevent the neural network search from falling to the minimum value. Then the BP neural network is used to search for the optimal solution among the smaller solutions, which makes the detection sample classification more accurate.

GA can classify and screen various groups and adjust and optimize populations by means of crossover and mutation according to the differences in their adaptability. Individuals with high fitness are more likely to survive. Through repeated iteration of the genetic operator, the individuals that meet the conditions are finally selected, that is, the approximate optimal solution.

According to the neural network structure to determine the length of the individual, the network ownership value is encoded in real time, as a set of chromosomes, that is

$$X = \{w_1, w_2, w_3, \cdots, w_n\} \tag{15}$$

where: $w_n$ is the weight value of the implicit layer's connection to the output layer.

To minimize the error between the actual value and the predicted value, the optimal individual is selected from the weight coding group by the genetic algorithm, and the weights of each layer of the neural network are initialized. After selecting the initial individual population, the neural network is tested using the training data set and the actual results are output. Here, the inverse sum of neural network error squares is used as an adaptation function.

$$SE = \frac{1}{n} \sum_{m=1}^{n} (y_i - x_i)^2 \tag{16}$$

$$f = \frac{1}{SE} \tag{17}$$

where: $n$ is the number of nodes; $x_i$, $y_i$ are the predicted output and expected output of the $i$th node, respectively.

Gene selection: According to the above-style matching degree training of individuals, using the roulette selection method and the principle of probability ratio selection, then each gene selected probability $P_i$ is as follows:

$$P = \frac{f(X_i)}{\sum\limits_{i=1}^{n} X_i} \tag{18}$$

Gene crossover: Use crossover operators to improve the individual coding structure and optimize the genes globally. By selecting genes $X_i$ and $X_k$ in the above formula, cross-operation is performed on the $j$th position of their chromosomes, namely:

$$\begin{cases} X_{kj} = (1+b)X_{kj} - X_{ij}b \\ X_{ij} = (1+b)X_{ij} - X_{kj}b \end{cases} \tag{19}$$

In the equation: $b$ is a constant, and the value range is 0–1.

Gene variation: In order to improve the local search ability of the algorithm and maintain the diversity of population, a gene point of the parent generation is replaced by an evenly distributed random number, making it more suitable for the current environment. The new gene points are:

$$X_{ij} = \begin{cases} X_{ij} + (X_{ij} - X_{max})f(g) & r \geq 0.5 \\ X_{ij} + (X_{min} - X_{ij})f(g) & r < 0.5 \end{cases} \tag{20}$$

$$f(g) = r_2(1 - \frac{g}{G_{max}}) \tag{21}$$

where: $X_{min}$, $X_{max}$ are the minimum and maximum values of the initial gene, $r_1$, $r_2$ is the value of the random number between 0 and 1; $G_{max}$ is the maximum number of evolutions; $g$ is the current number of iterations.

With each evolution of the population, the individual population adaptation increases until the individual meets the optimal iteration criteria, and the optimal solution is used as the corresponding weight value of the BP neural network in the default state. Genetic algorithms optimize the neural network flow as shown in Figure 6 below.

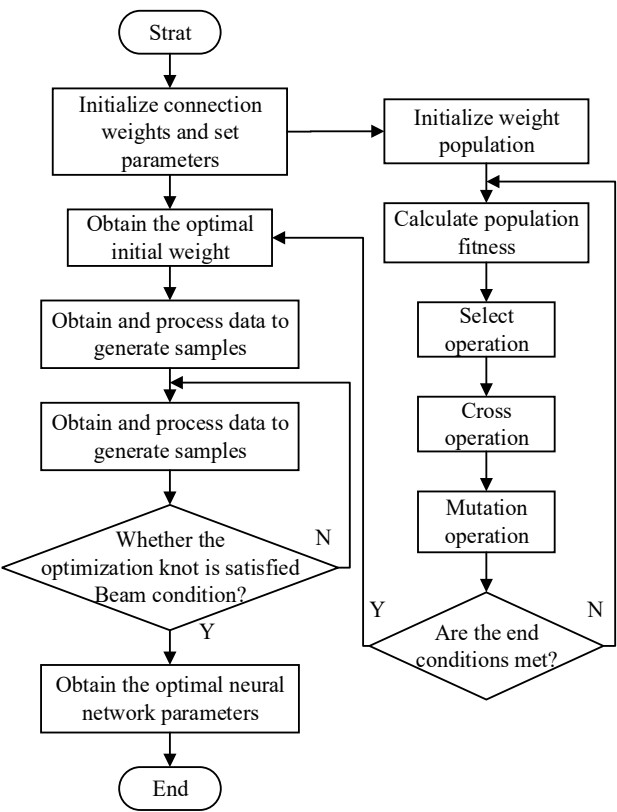

**Figure 6.** GA optimizes the BP neural network process.

### 4. Simulation and Case Studies

Based on the resulting integrated fault tree model, the integrated safety evaluation index system for electric vehicles is established as Figure 7.

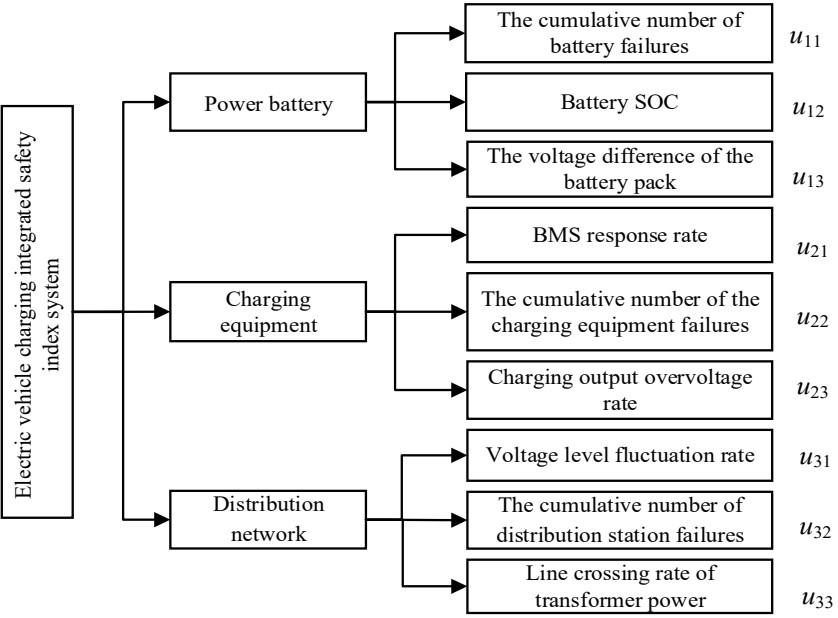

**Figure 7.** Integrated safety assessment index system for electric vehicles.

Ratings are scored for the previously established $V = \{v_1, v_2, v_3, v_4, v_5\}$ rating, which is rated as Table 2.

**Table 2.** Safety grade rating form.

| T | Level | Score |
|---|---|---|
| $100/t_1$ | Absolutely safety | >90 |
| $90/t_2$ | Safety | 80–90 |
| $80/t_3$ | General safety | 70–80 |
| $70/t_4$ | Dangerous | 60–70 |
| $60/t_5$ | Very dangerous | <60 |

According to the existing national standards, industry standards, enterprise standards and relevant expert advice and consultation, combined with the actual EV charging status, the data of charging failure caused by relevant factors are sorted out. At the same time, based on many data, the index threshold value triggering the EV charging safety is determined, and the classification table of the index safety grade of the EV integrated safety index system is established as Table 3.

**Table 3.** The safety rating of the integrated safety evaluation index for charging electric vehicles.

| Index | Hierarchy | | | | |
|---|---|---|---|---|---|
| | Absolutely Safety | Safety | General Safety | Dangerous | Very Dangerous |
| Cumulative number of battery failures (times) | 1 | 3 | 5 | 10 | 15 |
| Battery SOC (%) | 99 | 95 | 90 | 85 | 80 |
| Voltage difference of the battery pack (mV) | 10 | 50 | 100 | 200 | 500 |
| BMS response rate (%) | 99.8 | 98.5 | 97 | 95 | 90 |
| Cumulative number of charging equipment failures (times) | 0 | 2 | 4 | 5 | 8 |
| Charge output overvoltage rate (%) | 1.5 | 2.5 | 5 | 10 | 20 |
| Voltage level fluctuation rate of distribution network (%) | 0.3 | 1 | 2 | 3 | 5 |
| Cumulative number of distribution station failures (times) | 0 | 1 | 2 | 4 | 5 |
| Power over-line rate of distribution station (%) | 1 | 3 | 5 | 10 | 20 |

### 4.1. Simulation Analysis

Five hundred groups of EV charging data were randomly selected to form the training set, and 200 groups of sampling data were selected as the test, and nine indicators at the indicator layer were taken as the neural network input characteristic factors. One hundred groups of initialization weight populations were randomly selected and the fitness of each weight was calculated. The initialization mutation probability parameter was set as 0.25, the initialization crossover probability parameter was set as 0.4, and the maximum iteration number was set as 500. The optimal initial weights are obtained by iterative genetic operators. The node weights of the hidden layer and output layer of 500 groups of training data are used. The learning efficiency of the optimal weight group $\eta = 0.1$, momentum factor $A = 0.3$, after 100 iterations, the error value of the neural network meets the precision requirement. The changes of training accuracy and mean square deviation are shown in the Figure 8 below.

Figure 8 shows the accuracy changes of two kinds of neural network training. With the gradual increase of training times, its accuracy becomes higher and higher. It can be seen that after about 25 training sessions, the advantages of the GA_BP neural network become gradually obvious, and its accuracy gradually exceeds that of the BP neural network before improvement. Figure 9 shows the changing rule of the mean square error (MSE) of the BP neural network and the GA_BP neural network as the number of iterations increases. It can be seen from Figure 9 that, during the training process, the mean square error of the training group and the test group is approximately equal, and the mean square error of the GA_BP algorithm basically stays between 0.009 and 0.012, which is relatively small,

indicating that the accuracy of the network basically meets the calculation requirements. In comparison, the mean square error of the BP algorithm is relatively large, ranging from 0.060 to 0.070. In addition, it can be concluded that the training based on the GA_BP neural network has higher accuracy.

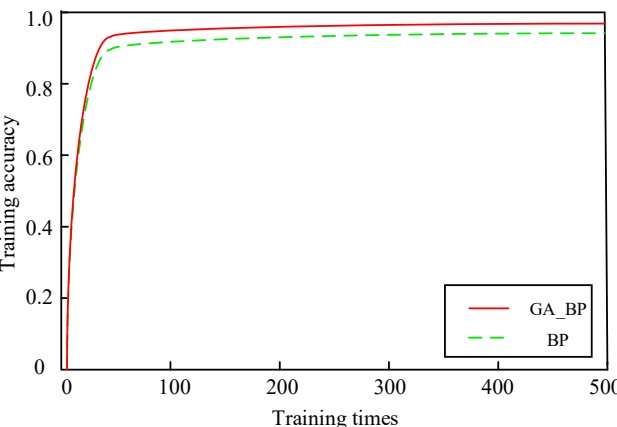

**Figure 8.** Comparison before and after neural network optimization.

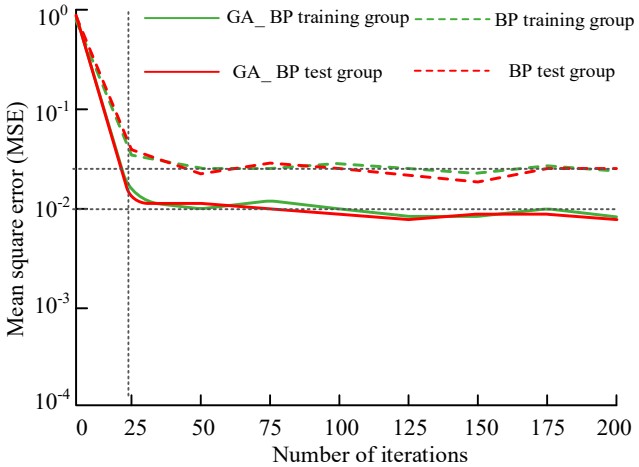

**Figure 9.** Change of mean square error before and after improvement of the BP neural network.

### 4.2. Case Study

Nanjing is one of the cities in China vigorously promoting electric vehicles. In recent years, the government has been strengthening the construction of the charging equipment infrastructure, and more users are now running electric vehicles. Therefore, the charging data of five groups of electric vehicles in a specific area in Nanjing in 2020 were selected for safety evaluation and analysis. The EVs in the five sets of data include EVs that are used shortly after purchase and EVs that have been driven for many years. The charging facilities include newly built charging stations and charging stations with long service life. In this way, the charging situation of the charging vehicles can be more comprehensively reflected. The comprehensive fuzzy evaluation method is used for data safety evaluation and analysis, and the obtained charging data is shown in Table 4.

**Table 4.** Five groups of data of electric vehicle charging in a specific area of Nanjing in 2020.

| Index | A | B | C | D | E |
|---|---|---|---|---|---|
| Cumulative number of battery failures (times) | 1 | 3 | 5 | 10 | 15 |
| Battery SOC (%) | 99 | 95 | 90 | 85 | 80 |
| Voltage difference of the battery pack (mV) | 10 | 50 | 100 | 200 | 500 |
| BMS response rate (%) | 99.5 | 98.3 | 92.6 | 97.4 | 95.7 |
| Cumulative number of charging equipment failures (times) | 0 | 2 | 5 | 3 | 4 |
| Charge output overvoltage rate (%) | 2.4 | 3.2 | 11 | 4.1 | 5.6 |
| Voltage level fluctuation rate of distribution network (%) | 0.7 | 5.4 | 6.3 | 1.4 | 2.1 |
| Cumulative number of distribution station failures (times) | 1 | 4 | 3 | 0 | 2 |
| Power over-line rate of distribution station (%) | 1.8 | 10.6 | 8.5 | 2.6 | 5.7 |

The two indicators of battery SOC and BMS response rate of the charging equipment are the larger the better index. The rest are the smaller the better index. The membership degree matrix is established according to the relevant membership function [31]:

$$R_1 = \begin{bmatrix} 0.453 & 0.276 & 0.246 & 0.025 & 0 \\ 0.354 & 0.458 & 0.161 & 0.027 & 0 \\ 0.245 & 0.387 & 0.343 & 0.025 & 0 \end{bmatrix} \quad (22)$$

$$R_2 = \begin{bmatrix} 0.615 & 0.306 & 0.079 & 0 & 0 \\ 0.378 & 0.504 & 0.109 & 0.009 & 0 \\ 0.265 & 0.464 & 0.108 & 0.094 & 0.069 \end{bmatrix} \quad (23)$$

$$R_3 = \begin{bmatrix} 0.223 & 0.487 & 0.164 & 0.126 & 0 \\ 0.106 & 0.558 & 0.209 & 0.105 & 0.022 \\ 0.195 & 0.253 & 0.369 & 0.183 & 0 \end{bmatrix} \quad (24)$$

According to the GA_BP neural network, the fuzzy indexes of power battery safety, charging equipment safety and distribution network safety are obtained. Taking A group as an example, the single-layer fuzzy evaluation matrix is calculated:

According to the GA_BP neural network one can obtain the fuzzy indicator of power battery safety $W_1$ = (0.614 0.264 0.122), the fuzzy indicator of the safety of charging equipment $W_2$ = (0.658 0.225 0.117), the fuzzy index of distribution network safety $W_3$ = (0.584 0.228 0.188). Taking A group as an example, the single-layer fuzzy evaluation matrix is calculated:

$$R = \begin{bmatrix} W_1 R_1 \\ W_2 R_2 \\ W_3 R_3 \end{bmatrix} = \begin{bmatrix} B_1 \\ B_2 \\ B_3 \end{bmatrix} = \begin{bmatrix} 0.4015 & 0.3376 & 0.2354 & 0.0255 & 0 \\ 0.5208 & 0.3690 & 0.0891 & 0.0130 & 0.0081 \\ 0.1911 & 0.4592 & 0.2126 & 0.1319 & 0.0050 \end{bmatrix} \quad (25)$$

Then according to Formula (13) one can find the electric vehicle integration evaluation degree evaluation sub-judgment matrix:

$$B = W \times R = \begin{bmatrix} 0.4073 & 0.3607 & 0.1940 & 0.0352 & 0.0027 \\ 0.4037 & 0.3589 & 0.1998 & 0.0351 & 0.0024 \\ 0.3891 & 0.3676 & 0.1978 & 0.0427 & 0.0028 \end{bmatrix} \quad (26)$$

Finally, based on the Formula (14) to get the charging all-in-one safety score value:

$$f = B \times T = \begin{bmatrix} 91.34 & 91.26 & 90.97 \end{bmatrix} \quad (27)$$

Similarly, based on the B–E group charging data calculated in accordance with the above steps, the resulting five sets of calculations are calculated, and the corresponding safety rating is expressed in Table 5 as follows.

**Table 5.** GA based evaluation results of the BP neural network.

| Constituencies | Power Battery | Charging Equipment | Distribution Network |
|:---:|:---:|:---:|:---:|
| A | 91.34 Absolute safety | 91.26 Absolute safety | 90.97 Absolute safety |
| B | 88.65 Safety | 85.46 Safety | 78.53 General safety |
| C | 77.48 General Safety | 71.67 General safety | 87.39 Safety |
| D | 75.34 General safety | 89.51 Safety | 91.02 Absolute safety |
| E | 69.38 Danger | 83.73 Safety | 88.62 Safety |

The results of each set of charging data based on the GA_BP neural network are visualized in radar graphics as Figure 10.

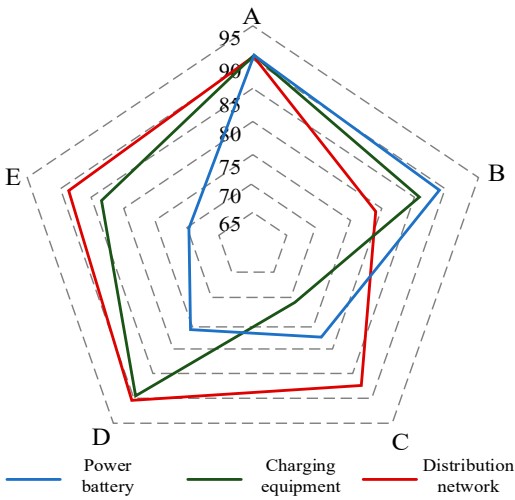

**Figure 10.** Evaluation results of five charging data in a region of Nanjing in 2020.

In the figure above, the closer the broken line is to the periphery, the higher is its safety evaluation degree. It can be seen intuitively from the figure that the charging data of group A indicates that the power battery, charging equipment, and power distribution platform area are all in the absolute safety state. In fact, it can be seen from the various index data that they are in a normal state. According to the license plate number and charging device ID of the vehicle, the vehicle has been running for less than two months and the charging station is also a newly built charging station. This region has a large substation capacity, and the distribution network operates steadily all year round, with demand response optimization in place. All the data show that the evaluation score is reasonable.

Then it can be seen that the power battery and charging equipment in group B are in a safe state. The distribution platform area is in a general safety state, and daily maintenance can be paid according to the scoring of the platform area. In the data of group C, the power battery is generally safe, the charging equipment is in the danger level, and the distribution platform area is safe. The power battery pack can be checked, and the corresponding charging equipment can be overhauled. The group D data charging equipment is in the general safe state, the charging equipment is in the safe state, and the distribution platform area is in absolute safety. According to the status of the power battery, it can be checked and maintained to prevent problems at a later period. Group E data show that the power battery is in a dangerous state, the charging equipment is generally safe, and the distribution platform area is safe. Therefore, replace and repair the battery string. Through the comparison and analysis of five groups of charging data, the safety scores and safety levels of electric vehicle charging power battery, charging equipment, and distribution platform area can be obtained. This can be used to repair or replace the equipment with low safety level, and the results of in-depth analysis are also in line with the actual situation, which proves the effectiveness of the proposed integrated safety evaluation model.

## 5. Conclusions

In this paper, the integrated safety state evaluation system of the vehicle–pile–grid was established by using a fault tree analysis method, a fuzzy comprehensive evaluation method, and a neural network algorithm. First, by analyzing the common faults of the power battery, the charging facilities, and the distribution network in the charging process, an integrated fault tree system was constructed to provide an analysis basis for fault diagnosis of electric vehicles. The corresponding evaluation indexes were obtained by sorting and summarizing the fault types. Based on these indexes, an integrated security state assessment system was constructed. The combination of the synthetic fuzzy evaluation method and the neural network algorithm increases the accuracy and reliability of the evaluation results. Finally, through simulation analysis, the results show that the evaluation effect based on the GA_BP neural network is better than the traditional BP neural network. Through a case study, the GA_BP neural network was used to evaluate the safety of the EV charging process in a specific area of Nanjing city, and the results proved the feasibility of the algorithm. The integrated security state assessment system proposed in this paper can provide a reference for the application of security assessment in other fields.

**Author Contributions:** Conceptualization, H.G. and B.Z.; methodology, H.G. and B.Z.; validation, H.G. and B.Z.; formal analysis, H.G. and B.Z.; investigation, H.G., B.Z. and L.S.; resources, H.G. and L.S.; data curation, H.G., L.S. and L.C.; writing—original draft preparation, B.Z. and L.C.; writing—review and editing, H.G. and B.Z.; visualization, H.G., B.Z. and L.S.; supervision, H.G. and L.C.; project administration, H.G., L.S. and B.Z.; funding acquisition, H.G. and L.S. All authors have read and agreed to the published version of the manuscript.

**Funding:** National Natural Science Foundation of China, No. 52077107, Early-warning analysis and operation and maintenance service of pile failure in electric vehicle charging process.

**Institutional Review Board Statement:** Not applicable.

**Informed Consent Statement:** Not applicable.

**Data Availability Statement:** Not applicable.

**Conflicts of Interest:** The authors declare no conflict of interest.

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
