# Peer review of "Evaluation of Electric Vehicle Integrated Charging Safety State Based on Fuzzy Neural Network"

_applsci, doi:10.3390/app12010461_

Round 1
Reviewer 1 Report
Summary:
This manuscript presents an integrated safety assessment method for electric vehicle charging safety based on fuzzy neural network. The topic is important to accurately evaluate the safety state of EV charging and lay the foundation for reducing charging safety accidents. However, there are some points in this manuscript that need to be further clarified. It is noted that this manuscript needs careful editing to avoid the obvious mistakes.
Major comments:
- On the Page 5, Line 205, “M and E represent intermediate events.” What is the difference between M and E and where does E appear in the text?
- According to Formula 5, the value of i ranges from 1 to n. Why does the value of i in Formula 8 range from 1 to m? Do these two i have different meanings?
3.On the Page 12, Line 348, “At the same time, based on a large number of data, the index threshold value triggering EV charging safety is determined.” Can you specifically introduce how the index threshold value triggering EV charging safety is determined?
Minor comments:
- In Figure 1, would it be better to replace “2. Influence of net on pile” with “2. Impact of power grid on pile” so that the text in Figure 1 can be consistent.
- On the Page 5, Line 205 and 206, “X is the base event.” Where does X appear in the text and what is R?
- According to Figure 5, it is R1, R2, ...Rn, but in Formula 12 is R1, R2 ,...Rm, is the number of rows in this matrix correct?
- Formula 13 has the same problem as Formula 12. Is the number of rows in the R matrix correct?
Reviewer 2 Report
Dear authors,
The topic introduced in your paper is interesting. The way it is presented is also correct. However, I would like you make some recommendations before publication:
- Abstract should include the main results, in a brief way.
- Including the structure of the paper at the end of the introduction part (“1. Introduction”) would add value to your paper. For instance: “The paper is organized as follows: section 2 presents the…, section 3…”
- I reckon that the introduction paragraph of section 2 (lines 97 to 107), section 2.1 and section 2.2 lack references that justify the presented statements.
- Section 4.2. states that the case study is a certain area of Nanjing. I recommend including the information about the selected case study that justifies its suitability for this paper: why was it selected? what are the features of the selected five groups of electric vehicles? What is the situation of Nanjing regarding electric vehicles? etc
